# National-Scale Orchard Mapping and Yield Estimation in Pakistan Using Object-Based Random Forest and Multisource Satellite Imagery

**DOI:** 10.3390/s25247468

**Published:** 2025-12-08

**Authors:** Ansar Ali, Ibrar ul Hassan Akhtar, Maisam Raza, Amjad Ali

**Affiliations:** SAR Wing, Pakistan Space and Upper Atmosphere Research Commission (SUPARCO), SUPARCO HQ, Islamabad Expressway, Islamabad P.O. Box 1271, Pakistan; ibrar.space@gmail.com (I.u.H.A.); raza88@hotmail.com (M.R.); amjadalee65@yahoo.com (A.A.)

**Keywords:** orchard delineation, Random Forest, OBIA, Google Earth Engine, multi-sensor satellite imagery, yield modeling, Pakistan

## Abstract

**Simple Summary:**

Fruit orchards are vital to Pakistan’s economy, yet no comprehensive or accurate record of their extent and productivity has previously existed. This study presents a new, cost-effective approach for mapping and monitoring orchards using both international and national satellite imagery. By integrating freely available Sentinel-2 data with high-resolution images from Pakistan’s PRSS-1 satellite, the research team produced detailed maps of citrus and mango orchards across major agro-ecological zones, achieving markedly improved boundary precision. Field surveys and statistical modeling linked satellite-derived vegetation indicators to actual fruit yields, enabling yield prediction directly from canopy reflectance and vigor. The results demonstrated strong predictive accuracy and regional scalability, confirming the method’s reliability for operational monitoring. This work establishes Pakistan’s first regionally validated geospatial system for orchard inventorying and yield estimation, offering a practical tool for precision farm management, export planning, and food-security decision-making, and setting the foundation for national-scale horticultural monitoring in other data-scarce regions.

**Abstract:**

Accurate geospatial inventories of fruit orchards are essential for precision horticulture and food security, yet Pakistan lacks consistent spatial datasets at district and tehsil levels. This study presents the first national-scale, object-based Random Forest (RF) framework for orchard delineation and yield estimation by integrating multi-temporal Sentinel-2 imagery on Google Earth Engine (GEE) with high-resolution Pakistan Remote Sensing Satellite-1 (PRSS-1) data. Among the tested classifiers, RF achieved the highest performance on Sentinel-2 data (Overall Accuracy (OA) = 79.0%, kappa (κ) = 0.78), outperforming Support Vector Machines (OA = 74.5%, κ = 0.74) and Gradient Boosting Decision Trees (OA = 73.8%, κ = 0.73), with statistical significance confirmed (McNemar’s χ^2^, *p* < 0.01). Integrating RF with Object-Based Image Analysis (OBIA) on PRSS-1 imagery further enhanced boundary precision (OA = 92.6%, κ = 0.89), increasing Producer’s and User’s accuracies to 90.4% and 91.5%, and increasing Intersection-over-Union (IoU) from 0.71 to 0.86 (*p* < 0.01). Regression-based yield modeling using field-observed data revealed that mean- and median vegetation index aggregations provided the most stable predictions (R^2^ = 0.77–0.79; RMSE = 72–105 kg tree^−1^), while extreme-value models showed higher errors (R^2^ = 0.46–0.56; RMSE > 560 kg tree^−1^). The resulting multisensory geospatial inventory of citrus and mango orchards establishes a scalable, transferable, and operationally viable framework for orchard mapping yield forecasting, and resource planning, demonstrating the strategic value of national satellite assets for food security monitoring in data-scarce regions.

## 1. Introduction

Pakistan ranks among the world’s leading producers of citrus and mango, yet it lacks a validated geospatial inventory of orchards—a critical data gap that limits yield forecasting, pest surveillance, and export competitiveness [1]. The absence of spatially explicit information on orchard extent, condition, and productivity constrains the transition from conventional to precision horticulture, where timely, location-specific data are vital for improving resilience against climate variability and sustaining rural livelihoods [2]. The transition remains challenging due to limited access to high-resolution geospatial data, inadequate digital infrastructure, fragmented landholdings, and the lack of institutional mechanisms to integrate remote sensing into routine agricultural monitoring [3].

Globally, advances in remote sensing and machine learning have enabled large-scale monitoring of croplands, plantations, and agroforestry systems. However, fruit orchards pose unique mapping challenges due to irregular geometries, mixed cropping patterns, and spectral similarity with surrounding vegetation [4]. While studies from China, Europe, and the United States have demonstrated progress in orchard delineation and yield estimation [5,6], these regions benefit from well-established ground-truth databases, open-access agricultural statistics, and continuous high-resolution Earth observation programs (e.g., Copernicus, Gaofen). In contrast, Pakistan lacks harmonized field survey datasets, validated orchard boundaries, and publicly accessible geospatial inventories, making large-scale calibration and validation of remote sensing models challenging.

The availability of Sentinel-2 and Pakistan’s PRSS-1 satellite now offers a cost-effective opportunity to bridge this gap. Sentinel-2 provides freely accessible multispectral data (10–20 m) suited for vegetation monitoring [7], whereas PRSS-1 delivers high-resolution (0.98–2.89 m) imagery acquired through national infrastructure, enabling detailed mapping at local to regional scales [8]. The synergistic use of these sensors allows integration of broad-scale spectral information with fine-grained spatial detail, crucial for large-scale orchard landscapes.

Accordingly, this study develops a regionally validated geospatial framework for orchard inventorying and yield estimation in Pakistan by integrating machine learning with multi-source satellite imagery. The framework comprises:(i)Object-based Random Forest (RF–OBIA) classification of citrus and mango orchards across major agro-ecological zones, and(ii)Non-destructive yield modeling (kg tree^−1^) using field-calibrated vegetation indices.

While the present work establishes a baseline national-scale orchard inventory, it also demonstrates the framework’s scalability for temporal monitoring using multi-year Sentinel-2 imagery (2019–2024), enabling periodic updates and yield trend analyses. This integrated approach—linking national satellite assets with advanced analytical workflows—provides a scientifically robust and operationally transferable methodology for precision horticultural monitoring, with strong implications for policy, food security, and sustainable agricultural management in data-scarce regions.

## 2. Materials and Methods

### 2.1. Study Area and Sampling Design

The study was conducted across two major agro-ecological zones of Punjab, Pakistan—Central Punjab Citrus Belt (31.50–32.80° N, 72.20–73.90° E), encompassing Sargodha and Mandi Bahauddin) and Southern Punjab Mango Belt (28.10–30.40° N, 70.30–71.90° E), including Multan, Khanewal, and Rahim Yar Khan districts—which together represent the country’s most intensive fruit-producing regions (Figure 1).

These zones differ substantially in climatic regimes, soil composition, and cropping systems, which collectively influence orchard spectral behavior and productivity [9]. The delineation of these zones follows the National Agro-Ecological Zoning Atlas of Pakistan (https://gaez.fao.org/datasets/hqfao::agro-ecological-zones-of-pakistan, accessed on 14 October 2025).

Central Punjab (Citrus Belt): Sub-humid subtropical climate; well-drained alluvial soils; canal irrigation; dominated by Kinnow mandarin (*Citrus reticulata*) within wheat–citrus–fodder rotations.Southern Punjab (Mango Belt): Arid to semi-arid climate; clay-loam to sandy clay-loam soils; canal/tubewell irrigation; dominated by *Mangifera indica* (Chaunsa, Sindhri, Langra, Dusehri) intercropped with cotton, wheat, and fodder.

These contrasting agro-ecological settings with significant biodiversity provide an ideal testbed for evaluating the robustness and transferability of the proposed Random Forest–OBIA framework. Key climatic and soil characteristics of these agro-ecological zones are summarized in Table 1.

### 2.2. Data Sources and Preprocessing

An overview of the methodological workflow integrating orchard delineation and yield regression modeling is shown in Figure 2.

#### 2.2.1. Satellite Data Acquisition and Preprocessing

A multi-sensor approach was adopted, combining Sentinel-2 MSI (10–20 m, Level-2A) and Pakistan Remote Sensing Satellite-1 (PRSS-1) imagery (0.98–2.89 m) as summarized in Table 2. Sentinel-2 data (2019–2024) were accessed via Google Earth Engine (GEE) for multi-temporal analysis of canopy vigor and yield trends. Preprocessing in GEE included Sen2Cor atmospheric correction, QA60 cloud masking, and temporal median compositing to minimize atmospheric noise [10].

PRSS-1 multispectral and panchromatic scenes (2022–2024) were acquired through SUPARCO’s Satellite Ground Station during peak fruiting periods (December–February for citrus; June–August for mango). All scenes underwent radiometric calibration to top-of-atmosphere reflectance and RPC-based orthorectification using DEM and ground control points ensuring geometric and radiometric consistency with Sentinel-2 imagery [8]. This integration enhanced geometric precision and spectral depth for canopy-based yield estimation.

#### 2.2.2. Field Sampling and Data Collection

Field surveys were conducted during 2022–2024 using a stratified random sampling approach designed to capture variability in orchard management, canopy structure, and agro-ecological conditions. The stratification was based on four key criteria: (i) crop type (citrus vs. mango), (ii) agro-ecological zone (Central Punjab vs. Southern Punjab), (iii) orchard size class (<5 ha, 5–10 ha, >10 ha), and (iv) management regime (irrigated vs. rainfed).

A total of 1524 GPS-referenced orchards were surveyed using a Trimble R1 differential GPS (sub-meter accuracy), including 812 citrus orchards and 712 mango orchards. Sampling coincided with peak fruiting seasons (December–February for citrus; June–August for mango) to ensure phenological consistency and maximum canopy vigor. All surveyed orchards were monoculture, uniformly managed commercial plantations (either citrus or mango) except for a few with heterogeneous tree species along their boundaries.

Attributes recorded in the field included canopy density, vigor, phenological stage, irrigation source, tree age, and health condition. These were cross-validated with PRSS-1 imagery for geometric alignment.

Field-measured fruit yield (kg tree^−1^) was collected from representative orchards and linked with spectral–structural indicators (NDVI, NDRE, texture metrics) to develop non-destructive regression models. After quality control, data were partitioned into 70% training and 30% validation subsets. Descriptive statistics of field-observed yield data across both agro-ecological zones are provided in Table 3, which served as the empirical foundation for defining orchard yield classes and calibrating yield prediction models.

#### 2.2.3. Machine Learning Classifiers Training and Validation

Three supervised classifiers—Random Forest (RF), Support Vector Machine (SVM), and Gradient Boosting Decision Trees (GBDT)—were implemented and evaluated on multi-temporal Sentinel-2 composites within Google Earth Engine (GEE). Model parameters were configured as follows:RF: ntree = 500; mtry = 3–5 (optimized by minimum out-of-bag error).SVM: Radial basis function (RBF) kernel; C and γ optimized via grid search.GBDT: Learning rate = 0.05; maximum tree depth = 5.

Performance was assessed using Overall Accuracy (OA), Kappa coefficient (κ), and Producer’s and User’s Accuracies derived from the independent validation dataset. RF achieved the highest accuracy (OA = 79%, κ = 0.78), outperforming SVM (74.5%, κ = 0.74) and GBDT (73.8%, κ = 0.73). Hence, RF was selected as the operational classifier for large-scale orchard mapping.

#### 2.2.4. Object-Based Boundary Refinement Using PRSS-1 Imagery

To enhance boundary precision, Object-Based Image Analysis (OBIA) was applied to PRSS-1 imagery. Multi-resolution segmentation using spectral, textural, and geometric features (mean bands, GLCM textures, shape index, compactness) generated homogeneous orchard objects.

Rule-based classification integrating vegetation indices (NDVI, NDRE) and brightness thresholds was used to discriminate orchards from croplands or settlements [11]. The OBIA-derived polygons were spatially aligned with Sentinel-2 RF outputs, producing a composite orchard inventory with substantially improved spatial fidelity. Validation against GPS field boundaries and visually interpreted PRSS-1 subsets confirmed accuracy gains.

#### 2.2.5. Model Accuracy Assessment and Cross-Validation

Classifier reliability was validated using the independent 30% field subset. Metrics included OA, κ, Producer’s and User’s Accuracies [12]. Special attention was given to minimizing commission/omission errors across mixed orchard mosaics, ensuring reproducibility for provincial-scale monitoring.

#### 2.2.6. Yield Regression Modeling and Evaluation

Field-observed orchard yields (kg tree^−1^) obtained during the ground campaigns (Section 2.2.2) were regressed against vegetation indices (NDVI, NDRE, SAVI, GNDVI, MCARI, RENDVI, NDMI, TNDVI) previously reported to correlate strongly with fruit tree productivity [13]. For each orchard polygon, mean, median, maximum, and minimum VI values were extracted to represent canopy heterogeneity. This non-destructive modeling approach allowed extrapolation of yield predictions to unmapped areas while maintaining calibration fidelity with field-observed yield.

Multiple Linear Regression (MLR) models were developed using the training dataset; performance was evaluated via R^2^, adjusted R^2^, and Standard Error of Estimate (SEE). Diagnostic checks (residuals, VIF, ANOVA) confirmed statistical significance (*p* < 0.001).

## 3. Results

### 3.1. Classifier Performance for Orchard Delineation

The supervised classifiers exhibited robust performance in delineating orchard extents across contrasting agro-ecological zones. Among the evaluated algorithms, the Random Forest (RF) classifier consistently outperformed Support Vector Machine (SVM) and Gradient Boosting Decision Trees (GBDT) when applied to pixel-based Sentinel-2 composites. RF achieved an overall accuracy (OA) of 79.0% (95% CI: 77.2–80.7%) and κ = 0.78, compared with 74.5% (κ = 0.74) for SVM and 73.8% (κ = 0.73) for GBDT (Figure 3a). The improvement was statistically significant (McNemar’s χ^2^ test, *p* < 0.01), confirming the superior generalization ability of RF under heterogeneous orchard canopy conditions.

Following RF classification, object-based boundary refinement using PRSS-1 imagery further enhanced delineation accuracy, improving boundary precision and polygon completeness (Figure 3b).

The enhanced performance of RF is attributed to its ensemble-based architecture, which minimizes overfitting, handles nonlinear interactions, and remains stable under spectral variability typical of mixed orchard mosaics [14]. Moreover, RF requires limited parameter tuning compared with GBDT, providing greater operational efficiency for large-scale mapping [15].

Feature importance analysis (Figure 4) highlighted the strong discriminatory contribution of the red-edge (Bands 5–6), near-infrared (Band 8), and shortwave-infrared (Band 11) regions, which are sensitive to canopy vigor, chlorophyll concentration, and structural moisture content. Temporal median compositing further stabilized spectral responses by minimizing phenological noise. These findings corroborate earlier research emphasizing the diagnostic utility of the red-edge–NIR domain for woody perennial crop mapping, with SWIR enhancing canopy moisture sensitivity [16].

### 3.2. Boundary Enhancement with OBIA and IoU Validation

The integration of object-based image analysis (OBIA) with Random Forest (RF) outputs substantially enhanced orchard boundary delineation, particularly in fragmented and mixed-use landscapes. High-resolution PRSS-1 imagery provided the geometric detail necessary for accurate object segmentation, while multi-resolution rule-based filtering minimized spectral confusion along orchard edges.

It is important to note that the OBIA refinement was applied to the complete mapped orchard inventory rather than limited field-sampled areas. The object-based segmentation and rule-based classification were implemented over the entire study extent to ensure geometric consistency and improve polygon coherence across all delineated orchard blocks. Field-sampled points were subsequently used only for independent validation and accuracy assessment.

Quantitatively, the OBIA-refined RF model achieved a marked performance gain, increasing Overall Accuracy (OA) from 79.0% to 92.6% (Δ = +13.6%, 95% CI: 10.1–13.9%) and the κ from 0.78 to 0.89. Producer’s Accuracy (PA) and User’s Accuracy (UA) also improved to 90.4% and 91.5%, respectively, compared with 77.5% and 80.2% for the baseline pixel-based RF classifier (Table 4). Intersection-over-Union (IoU) scores rose from 0.71 to 0.86 (*p* < 0.01, paired t-test), indicating closer geometric overlap between the classified and reference orchard boundaries. The most notable gains occurred in orchards smaller than 5 ha, where boundary fragmentation was greatest.

These improvements were further benchmarked against baseline classifiers in Section 3.4 to quantify statistical significance and boundary precision gains.

The performance improvement is primarily due to OBIA’s ability to incorporate spectral, geometric, and textural cues into coherent objects, suppressing intra-class spectral noise and reducing the “salt-and-pepper” effect typical of per-pixel classifiers [17]. The fusion of Sentinel-2’s multispectral variability with PRSS-1’s high spatial detail improved delineation of orchard edges, irrigation channels, and boundary vegetation. Temporal compositing further stabilized classification by mitigating phenological fluctuations [8].

Visual comparisons of delineation outputs further confirmed these quantitative improvements. In the citrus belt (Central Punjab), OBIA-refined boundaries closely followed true orchard geometry, exhibiting smoother parcel edges and higher IoU overlap with reference boundaries (Figure 5).

In the Mango Belt (Southern Punjab), OBIA refinement improved delineation of smallholder orchards characterized by irregular tree spacing and mixed canopy ages, reducing omission of marginal trees and enhancing boundary continuity (Figure 6). These refinements demonstrate the scalability of the RF–OBIA framework across diverse orchard systems.

### 3.3. Accuracy Assessment

Validation using 1500 independent field samples confirmed the robustness and transferability of the RF–OBIA framework across districts. The integrated approach maintained OA > 91% and κ = 0.88–0.90 across all validation sites (Table 5). Misclassification remained below 9%, primarily occurring at orchard margins or in mixed vegetation mosaics.

At the crop level, Producer’s Accuracy (PA) reached 96% for citrus and 94% for mango, indicating strong detection capability for true orchard pixels, while User’s Accuracy (UA) exceeded 92% for both crops, confirming the reliability of mapped orchard boundaries. These results underscore the framework’s stability across both smallholder and large-scale commercial orchards, validating its scalability for regional and national horticultural monitoring.

### 3.4. Benchmarking RF–OBIA Performance

Quantitative benchmarking reinforced the superiority of the integrated RF–OBIA framework relative to the baseline pixel-based RF classifier. The OBIA-enhanced model achieved markedly higher accuracy (OA = 92.6%, κ = 0.89) and boundary precision (85.3%) and temporal noise reduction improved from 5% to 15%. McNemar’s χ^2^ test (χ^2^ = 14.72, *p* < 0.001) confirmed the statistical significance of these gains (Table 6).

Class-level confusion matrices (Figure 7) confirmed higher class separability and reduced spectral confusion, while spatial benchmarking illustrated superior geometric consistency.

Spatial benchmarking outputs (Figure 8) further demonstrated the robustness and operational scalability of the RF–OBIA. The mapped orchard distribution (Figure 8a) closely matched provincial agricultural statistics (https://www.pbs.gov.pk/agriculture-statistics/, accessed on 14 October 2025) confirming the model’s ability to generalize across diverse landscape structures and management regimes, while the spatial distribution of IoU values (Figure 8b) indicate the degree of geometric agreement between classified and reference orchard boundaries. Higher IoU scores (>0.85) were observed in contiguous orchard blocks of Central Punjab and the commercial mango zones of Multan and Khanewal, whereas slightly lower values appear in fragmented smallholder areas with varied canopy patterns. These results make the RF–OBIA framework a reliable foundation for large-scale horticultural monitoring and yield modeling.

### 3.5. Strategies for Yield Modeling and Regression Analysis

#### 3.5.1. Effect of Vegetation Index Aggregation on Yield Prediction

Regression analysis revealed that vegetation-index (VI) aggregation strategy significantly influenced yield-prediction accuracy. Mean- and median-based models achieved the highest explanatory power (R^2^ = 0.77–0.79) with residual errors of 72–105 kg tree^−1^, whereas maximum- and minimum-based models performed poorly (R^2^ = 0.46–0.56; errors > 560 kg tree^−1^) (Table 7).

Mean/median aggregation outperformed extremes by suppressing local anomalies (canopy gaps, shadowing, sensor noise), thus producing more stable, ecologically representative yield estimates [18]. Scatterplots of observed versus predicted yields clustered tightly around the 1:1 line for mean and median models indicating minimal bias (Figure 9).

#### 3.5.2. Correlation Analysis of Vegetation Indices

Pearson correlation analysis (Figure 10) confirmed strong positive relationships between yield and NDVI (r = 0.79), SAVI (r = 0.77), and TNDVI (r = 0.75) under mean and median aggregations. Red-edge indices (NDRE, RENDVI) based aggregation models exhibited moderate correlations, while MCARI and NDMI models were weakly associated due to their sensitivity to structural and moisture anomalies [19].

#### 3.5.3. Spatial Distribution of Predicted Yield

Spatially explicit orchard-level yield maps derived from the RF–OBIA delineation revealed pronounced spatial heterogeneity across Punjab’s fruit belts, highlighting the influence of agro-climatic gradients, management intensity, and canopy vigor on orchard productivity.

In the Citrus Belt (Central Punjab), predicted yields ranged from 175 to 2060 kg tree^−1^ (Figure 11). High-yielding orchards (≥1500 kg tree^−1^; Classes A–B) were concentrated in Sargodha, Bhalwal, Kot Momin– areas known for fertile alluvial soils, well-managed irrigation, and dense vigorous canopies. Moderately yielding orchards (550–1000 kg tree^−1^; Class C) characterized Silanwali, Bhera, and Malakwal reflecting variable management intensity and mid-aged canopies, while low-yield orchards (<500 kg tree^−1^; Class D) occurred primarily in Shahpur, Sahiwal, Lalian and fragmented smallholder plots exhibiting canopy gaps and senescent growth. The spatial gradient clearly delineates management-dependent productivity zones, reaffirming the model’s ability to capture fine-scale horticultural variability.

In the Mango Belt (Southern Punjab), predicted yields ranged from 173 and 2700 kg tree^−1^ (Figure 12). High yields (≥2000 kg tree^−1^) concentrated in Multan and Khanewal, reflected mature canopies and reliable irrigation. Moderate yields (800–1500 kg tree^−1^) dominated Rahim Yar Khan, where mixed-age orchards and seasonal water scarcity affected canopy vigor. Lower-yielding orchards (<500 kg tree^−1^) occurred along sandy loam tracts prone to moisture stress and sparse canopy cover.

#### 3.5.4. Error Diagnostics and Model Stability

Boxplot analysis (Figure 13) summarizes model stability across aggregation strategies. Mean- and median-based models exhibited compact interquartile ranges and minimal outliers, demonstrating consistent performance. Conversely, maximum- and minimum-based models produced wider yield distributions, indicating higher sensitivity to local spectral anomalies and illumination effects.

Residual diagnostics (Figure 14) confirmed these findings, with mean- and median-based models exhibiting residuals tightly clustered around the zero-error line and within ±2 standard errors of prediction, indicating strong model stability and minimal heteroscedasticity. In contrast, extreme-value models (maximum and minimum) showed higher residual variance and reduced consistency across orchards.

A slight underestimation trend was observed for high-yield orchards, which can be attributed to canopy saturation effects in NDVI-based indices, non-linear fruit–foliage relationships, and shadow interference that dampen spectral sensitivity under dense canopies [20,21]. Additional structural factors—such as planting density, pruning intensity, and soil fertility gradients—may also act as potential contributors to this bias [22]. Collectively, these results reaffirm that mean and median aggregation strategies yield the most stable, transferable, and ecologically consistent framework for orchard-scale yield estimation.

These integrated results establish a validated, operational framework for orchard delineation and yield prediction, forming the empirical basis for subsequent discussion on scalability, accuracy limitations, and policy relevance.

## 4. Discussion

Accurate mapping of perennial fruit orchards has long been constrained in South Asia by fragmented landholdings, mixed cropping patterns, and spectral confusion with surrounding vegetation. This study demonstrates that combining multisource satellite imagery—Sentinel-2 for spectral depth and PRSS-1 for spatial precision—within an object-based Random Forest framework effectively overcomes these challenges. The RF–OBIA integration substantially enhanced spatial coherence and thematic accuracy, underscoring the value of merging spectral richness with fine-scale segmentation for delineating varied orchard landscapes.

These findings reaffirm that data fusion across spatial and spectral domains is indispensable for perennial crop monitoring in regions where field data and cadastral records are limited. The methodological design aligns with global advances in object-based remote sensing, where machine learning integrated with high-resolution imagery has proven effective for precision agriculture and plantation mapping [23,24]. However, this work goes further by operationalizing such integration at a national scale using Pakistan’s own satellite infrastructure—demonstrating how indigenous space assets can underpin agricultural intelligence and policy support systems.

The yield-modeling component illustrates that aggregation strategy of vegetation indices (VIs) critically determines prediction reliability. Mean and median aggregations offered greater robustness than extreme-value measures, capturing canopy-level reflectance without amplifying local anomalies caused by shadowing or structural heterogeneity. These outcomes align with studies emphasizing the importance of aggregated spectral metrics for stabilizing yield predictions in orchard systems [24,25]. Nonetheless, a modest underestimation of high-yield classes suggests the presence of spectral saturation in broadband indices such as NDVI, signaling the need for complementary structural or hyperspectral observations to better capture canopy density and fruit load variations.

At the application level, the developed RF–OBIA framework provides a replicable foundation for precision horticulture and food-security monitoring. Its demonstrated scalability within the Google Earth Engine environment enables near-real-time national assessments, supporting decisions related to resource allocation, yield forecasting, and export planning. When integrated with ancillary datasets on soil, irrigation, and climate, such systems can evolve into operational geospatial decision-support platforms for climate-resilient orchard management—enabling early warning of yield declines, drought stress, and pest or disease outbreaks.

Despite these advances, some limitations merit consideration. The regression models rely on optical data and may be less effective under persistent cloud cover or varying illumination conditions. Furthermore, spectral-only indicators cannot fully capture the physiological complexity of canopy productivity [26]. Future work should therefore focus on (i) fusing optical, radar, and hyperspectral data to improve all-weather and structural sensitivity; (ii) leveraging deep learning architectures—such as convolutional and transformer models—to enhance temporal generalization; and (iii) incorporating transfer learning and crowdsourced data to extend the framework’s applicability across ecological and geographic contexts.

In summary, this research provides Pakistan’s first operational geospatial framework for mapping and yield estimation of fruit orchards, linking national satellite capabilities with machine learning and field calibration. Beyond its national relevance, the study offers a transferable methodology for perennial crop monitoring across the Global South—where data scarcity, smallholder dominance, and environmental variability remain major constraints. The integration of multi-sensor imagery, ensemble learning, and cloud-based analytics marks a decisive step toward data-driven precision horticulture and sustainable food-system transformation.

## 5. Conclusions

This study establishes a scientifically rigorous and operationally scalable framework for orchard mapping and yield estimation by integrating medium-resolution Sentinel-2 imagery with high-resolution PRSS-1 data through an object-based Random Forest (RF–OBIA) approach. The fusion of spectral and spatial domains markedly improved orchard boundary precision and thematic accuracy, achieving over 90% classification accuracy with strong correspondence to field-observed data across contrasting agro-ecological zones.

By demonstrating that mean- and median-aggregated vegetation indices provide the most stable and ecologically representative yield predictions, the research underscores the importance of aggregation strategy in mitigating spectral noise and improving model reliability. These findings collectively confirm that multi-sensor integration and ensemble learning can overcome the long-standing limitations of fragmented datasets, mixed cropping patterns, and data scarcity that have constrained horticultural monitoring in Pakistan.

Beyond its technical contributions, this work represents Pakistan’s first regionally validated, multi-sensor geospatial framework for fruit orchard inventorying—implemented entirely within a cloud-computing environment (Google Earth Engine) to ensure reproducibility, cost-efficiency, and scalability. The approach provides an operational pathway toward national-level monitoring of orchard dynamics, supporting data-driven decision-making for resource allocation, export planning, and climate resilience in the horticultural sector.

Future research should expand this framework through the integration of radar and hyperspectral datasets, advanced deep learning architectures, and agro-climatic modeling to enhance yield prediction under diverse environmental and management conditions. Such advancements will further consolidate the role of national satellite assets and artificial intelligence in building a precision horticulture ecosystem that strengthens food security and sustainable agricultural management across Pakistan and similar data-limited regions globally.

## Figures and Tables

**Figure 1 sensors-25-07468-f001:**
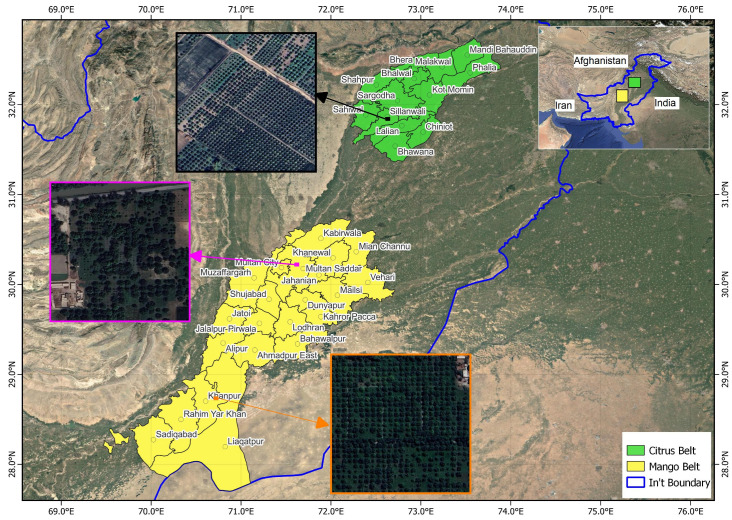
Geographic location of study regions: Green indicate the Citrus-producing belt in northeastern Punjab (Sargodha and Mandi Bahauddin), and yellow the Mango-producing belt in southwestern Punjab (Multan, Khanewal, Rahim Yar Khan).

**Figure 2 sensors-25-07468-f002:**
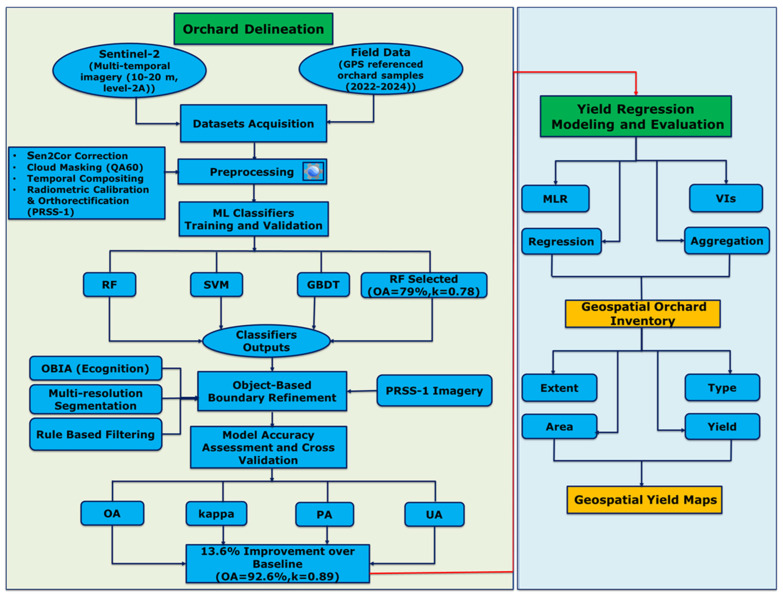
Methodological workflow illustrating the dual components of the study: (**left**) fruit orchard delineation and (**right**) yield modeling.

**Figure 3 sensors-25-07468-f003:**
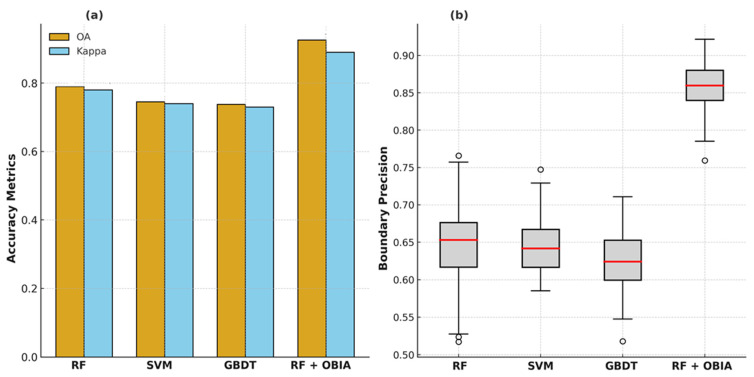
Classification performance of machine learning models for orchard delineation: (**a**) Comparison of Overall Accuracy (OA) and Kappa coefficients for Random Forest (RF), Support Vector Machine (SVM), Gradient Boosting Decision Trees (GBDT), and the integrated RF + Object-Based Image Analysis (RF+OBIA); (**b**) Boxplots showing variability in classifier performance across validation samples.

**Figure 4 sensors-25-07468-f004:**
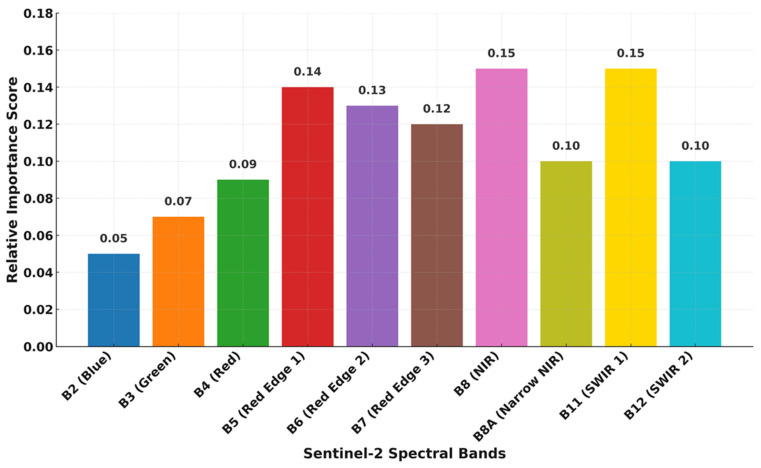
Variable importance of Sentinel-2 spectral bands in Random Forest classification for orchard delineation. Importance values represent the mean relative decrease in accuracy for each band.

**Figure 5 sensors-25-07468-f005:**
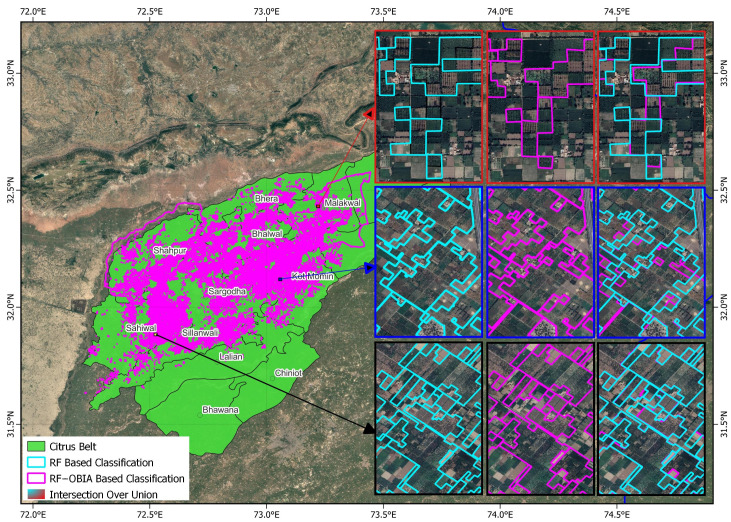
Orchard delineation results for the citrus belt (Central Punjab). Cyan outlines represent RF classification; magenta outlines show OBIA-enhanced RF results; Cyan—magenta composite indicate Intersection-over-Union (IoU) agreement with reference boundaries.

**Figure 6 sensors-25-07468-f006:**
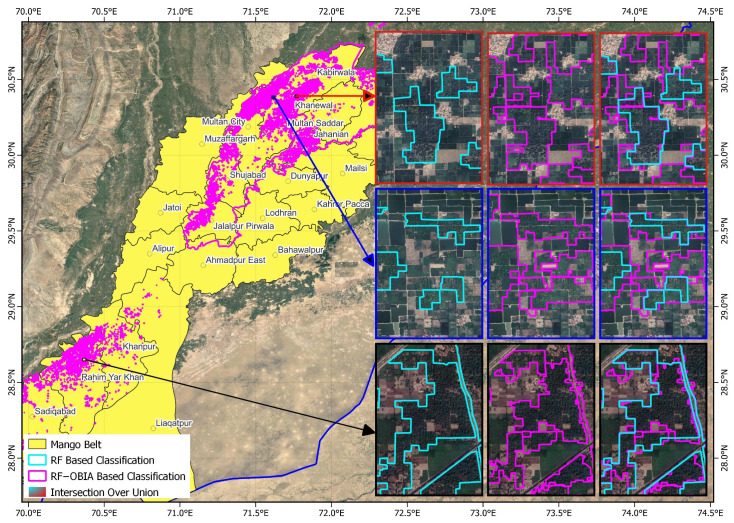
Orchard delineation results for the mango belt (South Punjab). Cyan outlines represent RF classification; magenta outlines show OBIA-enhanced RF results; Cyan—magenta composite indicate Intersection-over-Union (IoU) agreement with reference boundaries.

**Figure 7 sensors-25-07468-f007:**
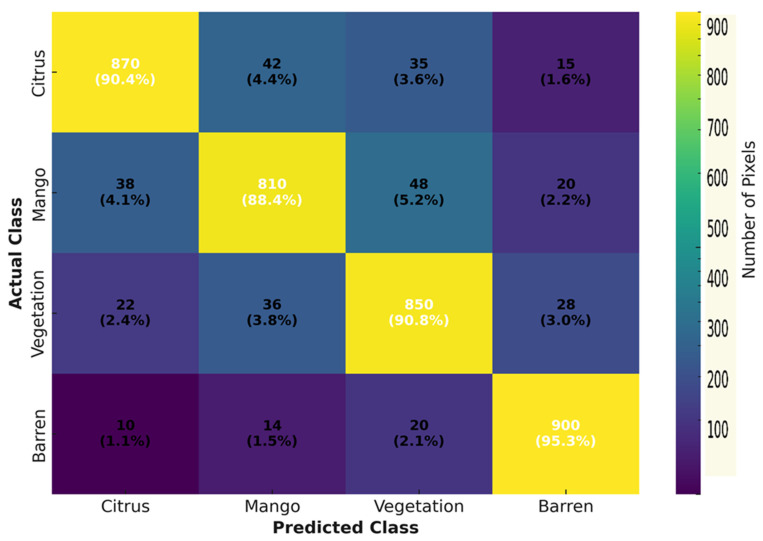
Confusion matrix showing class-level accuracy of RF–OBIA versus baseline RF classification.

**Figure 8 sensors-25-07468-f008:**
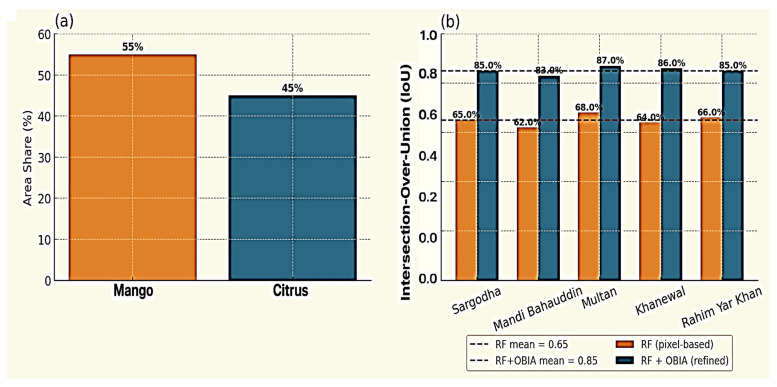
Benchmarking of RF–OBIA classification: (**a**) Distribution of orchard types based on validated inventory, (**b**) Spatial distribution of Intersection-over-Union (IoU) scores showing agreement between classified and reference orchard boundaries.

**Figure 9 sensors-25-07468-f009:**
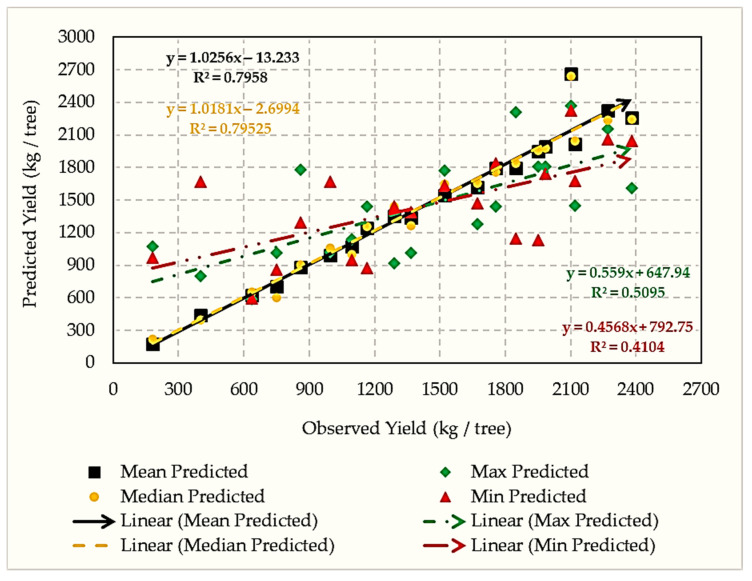
Observed versus predicted fruit yield under different vegetation index aggregation strategies (mean, median, maximum, and minimum).

**Figure 10 sensors-25-07468-f010:**
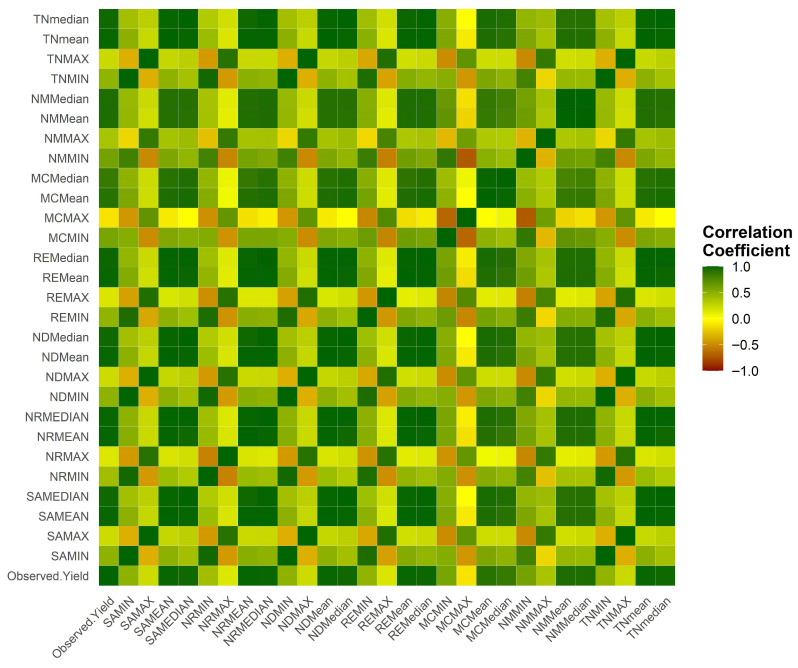
Pearson correlation matrix showing relationships among vegetation indices (NDVI, SAVI, TNDVI, NDRE, RENDVI, MCARI, and NDMI) based aggregation models and field-measured fruit yield. Green, yellow and brown to red colors denote positive (strong), weak or neutral and negative correlations, respectively.

**Figure 11 sensors-25-07468-f011:**
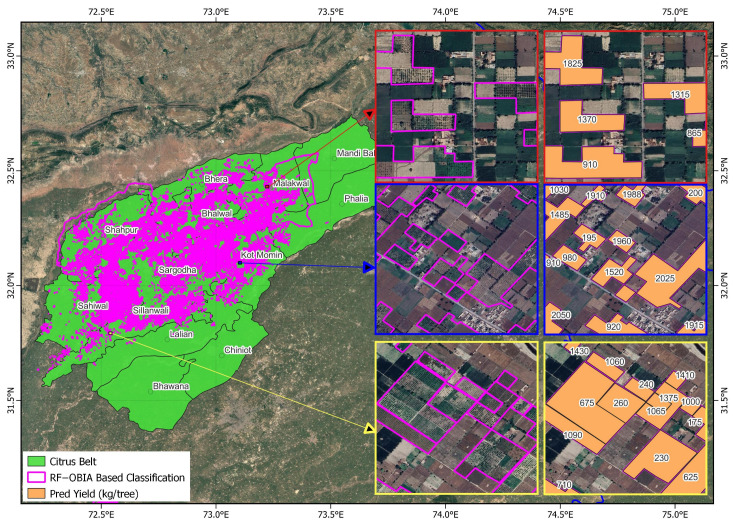
Spatial distribution of predicted citrus orchards yields (kg tree^−1^) Central Punjab derived from regression modeling using the RF–OBIA framework.

**Figure 12 sensors-25-07468-f012:**
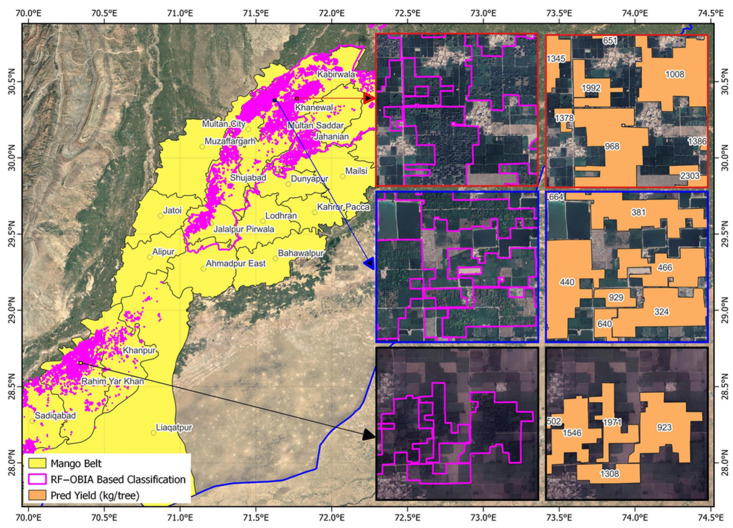
Spatial distribution of predicted mango orchards yields (kg tree^−1^) in Southern Punjab derived from regression modeling using the RF–OBIA framework.

**Figure 13 sensors-25-07468-f013:**
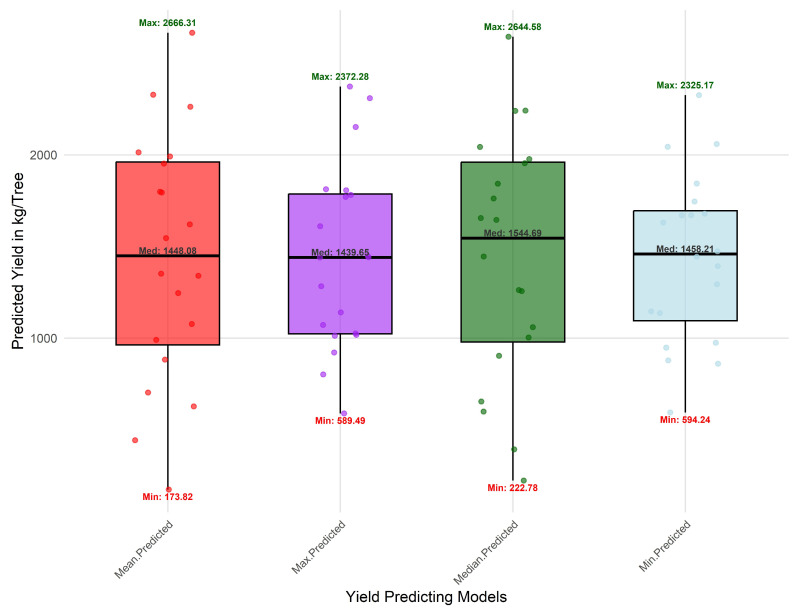
Boxplots showing distribution of predicted orchard yields (kg tree^−1^) across regression models using mean, median, maximum, and minimum aggregation strategies.

**Figure 14 sensors-25-07468-f014:**
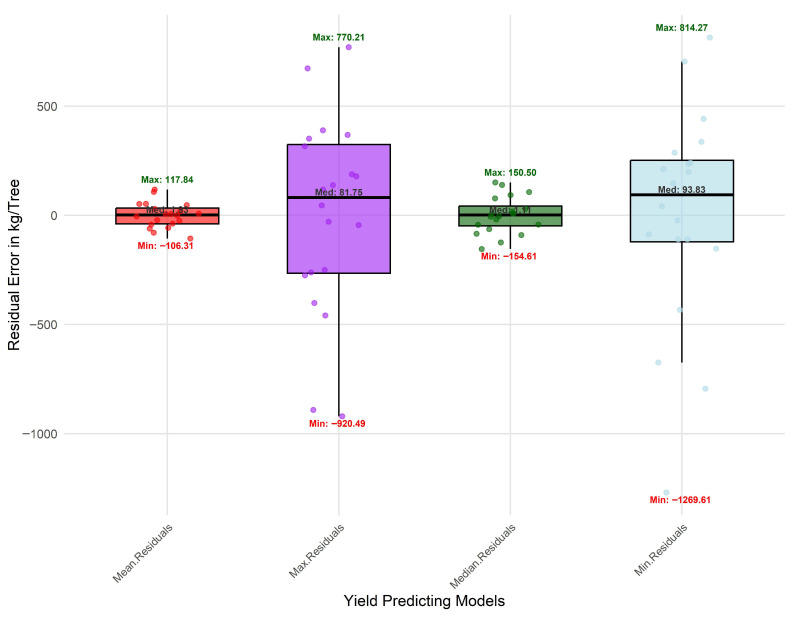
Residual diagnostic plots of regression models for orchard yield prediction under different vegetation index aggregation strategies (mean, median, maximum, and minimum).

**Table 1 sensors-25-07468-t001:** Agro-climatic characteristics of major orchard zones of Pakistan.

Zone	Dominant Fruit Crop	Climate Type	Mean Annual Temperature (°C)	Annual Rainfall (mm)	Dominant Soil Type	Cropping System Characteristics
Central Punjab (Citrus Belt)	Kinnow mandarin (*Citrus reticulata*), orange, lemon	Sub-humid to semi-arid subtropical	10–45	350–500	well-drained alluvial sandy-loam	Canal-irrigated integrated with wheat–citrus–fodder rotations
Southern Punjab (Mango Belt)	Mango (*Mangifera indica*)—Chaunsa, Sindhri, Langra, Dusehri	Arid to semi-arid subtropical	15–48	150–300	Clay-loam to sandy clay-loam alluvial soils	Canal and tubewell-irrigated orchards intercropped with cotton, wheat, and fodder crops

**Table 2 sensors-25-07468-t002:** Satellite datasets used for orchard delineation and yield estimation.

Sensor/Platform	Spatial Resolution	Temporal Coverage	Preprocessing Steps	Purpose in Study
Sentinel-2 MSI (ESA, Level-2A)	10–20 m	2019–2024 growing seasons (multi-temporal composites)	Sen2Cor atmospheric correction, QA60 cloud masking, temporal median compositing	Orchard classification, vegetation index derivation, yield modeling
PRSS-1 (Pakistan Re-mote Sensing Satel-lite-1)	0.98–2.98 m	2022–2024 (peak fruiting stages)	Radiometric calibration to reflectance, RPC-based orthorectification using DEM and GCPs	Boundary refinement, object-based segmentation in vast orchard landscapes

**Table 3 sensors-25-07468-t003:** Descriptive statistics of field-observed yield data across agroecological zones, orchard size classes, and management regimes.

Crop	Agro-Ecological Zone	Class	Observed Yield Range (kg tree^−1^)	Mean ± SD (kg tree^−1^)	Sample Count (n)	Description of Orchard Condition
Citrus	Central Punjab	A	1500–2300	1675 ± 110	210	Large, export-grade orchards with full canopy closure and uniform high fruit load
		B	1000–1480	1230 ± 95	240	Mature, healthy orchards with dense foliage and considerable fruit load
		C	550–950	720 ± 85	200	Middle-aged orchards with mixed canopy vigor and moderate fruit load
		D	150–500	340 ± 75	162	Senescent orchards with sparse canopy and low fruit density
Mango	Southern Punjab	A	2000–2700	2320 ± 135	180	Large, export-grade orchards with full canopy closure and uniform high fruit load
		B	1500–1950	1700 ± 115	210	Mature, vigorous orchards with dense canopy and considerable fruit load
		C	800–1450	1100 ± 120	185	Middle-aged orchards with mixed canopy vigor and moderate fruit load
		D	170–750	410 ± 95	137	Senescent orchards with sparse canopy and low fruit density

**Table 4 sensors-25-07468-t004:** Performance comparison of pixel-based and OBIA-enhanced classifiers for orchard delineation.

Classifier & Method	OA (%)	(κ)	PA (%)	UA (%)	ΔOA (%)	Δκ	IoU
RF (Pixel-based, Sentinel-2)	79.0	0.78	77.5	80.2	–	–	0.71
SVM (Pixel-based, Sentinel-2)	74.5	0.74	72.8	75.6	–4.5	–0.04	0.68
GBDT (Pixel-based, Sentinel-2)	73.8	0.73	71.9	74.1	–5.2	–0.05	0.67
RF—OBIA (Sentinel-2 + PRSS-1)	92.6	0.89	90.4	91.5	+13.6	+0.11	0.86

**Table 5 sensors-25-07468-t005:** Regional validation metrics for RF–OBIA classification of citrus and mango orchards across Punjab, Pakistan.

Region	Orchard Type	OA (%)	Kappa	Misclassification (%) **	PA (%) *	UA (%)
Sargodha	Citrus	92.3	0.89	7.7	96	93
Mandi Bahauddin	Citrus	91.8	0.88	8.2	95	92
Multan	Mango	93.0	0.90	7.0	94	93
Khanewal	Mango	92.6	0.89	7.4	94	92
Rahim Yar Khan	Mango	91.9	0.88	8.1	93	92

* Metrics derived using independent validation subsets. ** Errors primarily occurred at orchard margins or in mixed-crop mosaics.

**Table 6 sensors-25-07468-t006:** Benchmark comparison of OBIA—enhanced RF against baseline pixel-based RF classification.

Method	OA (%)	Kappa	Boundary Precision (%)	Temporal Noise Reduction (%)
RF (Pixel-based)	79.0	0.78	65.0	5.0
RF–OBIA	92.6	0.89	85.3	15.0

**Table 7 sensors-25-07468-t007:** Performance of regression models under different VI aggregation strategies for orchard yield prediction.

Aggregation Strategy	R^2^	Adjusted R^2^	RSE (kg tree^−1^) *
Mean	0.79	0.77	72.7
Median	0.78	0.78	76.4
Max	0.55	0.29	568.8
Min	0.46	0.14	626.6

All models significant at *p* < 0.001. * Residual Standard Error (RSE) represents model prediction deviation expressed in kg tree^−1^.

## Data Availability

The datasets generated and analyzed during the current study are available from the corresponding author upon reasonable request. Satellite datasets used (Sentinel-2 and PRSS-1) are publicly accessible through the Google Earth Engine (GEE) platform and SUPARCO’s Satellite Ground Segment, respectively. Derived geospatial layers and analytical scripts supporting the findings of this study can also be shared for academic and non-commercial use upon request.

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
