# Peer review of "National-Scale Orchard Mapping and Yield Estimation in Pakistan Using Object-Based Random Forest and Multisource Satellite Imagery"

_sensors, 2025, doi:10.3390/s25247468_

Round 1

Reviewer 1 Report

Comments and Suggestions for Authors

This study proposes a remote sensing monitoring framework for orchards at the national scale of Pakistan, integrating multi-source satellite data (Sentinel-2 and PRSS-1) with machine learning and object-based image analysis (OBIA) methods. The methodology is well-designed and holds significant reference value for precision horticulture and agricultural policy in Pakistan as well as in other data-scarce regions. It is recommended for acceptance after minor revisions.

  1. In the current manuscript version, the image resolution (e.g., figure 7) is relatively low, with issues such as blurry text labels, overly dense information, undersized fonts, and unclear legends. These problems are particularly evident when viewing the PDF in an enlarged format, where details become difficult to discern. It is essential to adjust the fonts and contrast to improve readability. The images must be made clearer; otherwise, the manuscript will not meet the publication standards. The attached image serves as an example of the quality observed on our end - basically, every image is unclear, and the words are difficult to read!
  2. According to the MDPI Template guidelines, it is recommended to format tables using the "three-line table" style.
  3. It is recommended to include a description of the radiometric calibration and orthorectification processes for the PRSS-1 data to ensure consistency with the Sentinel-2 data.
  4. In the discussion section, it is advisable to further elaborate on the generalizability and transferability of the proposed method to other data-scarce regions, thereby enhancing the study's international relevance.

This study demonstrates outstanding performance in methodological innovation, national-scale application, and policy relevance, with solid results. However, certain sections require further refinement in terms of methodological details, the quality of figure and table presentations, and the depth of result discussions. It is recommended that the authors revise the manuscript according to the above comments before it can be accepted.

Reviewer 2 Report

Comments and Suggestions for Authors

I think the authors aim to tackle a very important and ambitious topic: building a national-scale orchard monitoring framework. However, I felt the manuscript is too broad in some places and missing detail in others. For example it is unknown the characteristics of the agroecological zones or agroclimatic zones. In general, I recommend that authors balance the general results they are presenting without neglecting the specific details that support their results. The methods are not fully clear (especially sampling and agroecological context), figures/tables are hard to read, and the results are presented with lots of numbers but not enough interpretation. Also, the discussion sometimes claims more than what the results show. I recommend the following changes to improve your article:

MATERIALS AND METHODS

Figures 1 and 2:

  • Currently blurry and not readable.
  • Increase resolution, enlarge text and labels, and make the scale clearer.
  • Ensure that geographic names (districts, tehsils) are visible and legible.

Agro-climatic information:

  • The authors emphasize the importance of agro-climatic variation, but no details (temperature ranges, rainfall, soils, agro-ecological classification) are provided.
  • Please include a table or short description of climate and soil conditions.

Table 1: Not cited in the text at first mention.

Sampling method section 2.2.2:

  • The criteria used for stratified sampling are not specified.
  • Clarify what “management regimes” were considered (e.g., irrigated vs rainfed, conventional vs intensive).
  • Provide details on how >1,500 GPS points were distributed (by crop type, region, orchard size).
  • Explain whether sampling was balanced between citrus and mango, and between smallholder vs large-scale orchards.

Figure 2 (workflow): Make sure the steps shown in the figure must match exactly the methodological steps in the text.

Agro-ecological zones: The manuscript mentions zones in general but does not define them. Please describe the agro-ecological zones included, with their main characteristics (climate, soils, cropping systems and the characteristics of the cropping systems).

RESULTS

Figure 3: Abbreviations in panel (a) are not explained. Add definitions in the figure caption. Figure 3b is not explicitly cited or discussed in the text.

Figures 3–9 in general: The labels are too small, and figures are low resolution.

Line 213 mentions orchards under 5 ha but no details are provided of the type of cropping systems or the size of them. Clarify: are these monocultures or mixed orchards? Were they all under similar management? Were phenological stages the same? (This should be included in the methodology to make the results understandable.). This detail is important because the main claim is improved boundary delineation in fragmented orchards.

Table 2: Not cited when first presented.

Level of contextual detail: Results are broad and heavy on statistics, the absence of some details like for example, ecological zone characteristics, and types of faming systems, phenological stages, agroclimatic information etc.  makes it difficult to assess the robustness and generalizability of the findings. This is critical given the stated objective of improving boundary delineation across diverse agro-ecological zones. Adding this would help interpret whether the framework is robust across different realities.

DISCUSSION

The statement “these advances establish a nationally grounded geospatial approach for precision horticulture in a context where official orchard inventories have been absent” requires a supporting reference.

Scale of discussion vs results: The discussion makes claims about benefits at farm, regional, and national scales. However, the results presented are at regional scale only. Either, adjust the discussion to match results, or provide additional evidence/results at farm and national scales.

Yield underestimation: The study observes underestimation in high-yield orchards but does not analyze why. Please expand on structural factors that could explain this bias.

Reviewer 3 Report

Comments and Suggestions for Authors

Dear authors, 

It was pleasure to read this research and give you some ideas to improve it 

National-Scale Orchard Monitoring in Pakistan through 2 Machine Learning and Remote Sensing

The main research deals with the inventory of orchards and the inventory as a whole concept. The inventory was improved introducing the OBIA data, but some field data is missed when it refers to the yield assessment, which must be clarify.

28-46 Here some acronyms should be described (ML, OA, IoU)

53-56 Here must be added precise arguments and explanation what makes difficulties for the transition from conventional to precise farming.

62-67 Which kind data in Pakistan are limited and which are not limited in China, EU?

67 The acronym PRSS-2, what is that?

73-79 The research was focused on monitoring or inventory only. Th monitoring process implies time-record.

83-96 The study area is descried in very general way, there is no geographical coordinates of each district, the biodiversity. Nothing in regard the sampling design is described, just you intend to show some pictures.

106 The workflow resolution must be improved and send back, at this stage is too difficult to visualize the terms.

123-132 Then you should explain how many of sampling points were gathered, including the diverse orchard types and only of citrus and mango. So, at this point the partitioning between 70 and 30% does not have any meaning yet.

133-144 As the clear methodology, is not described the details, such as the supervised ML classifiers. The basis for 70 and 30 % partitioning. The model parameters to be calibrated.

145-153 The refinement was developed for the sampling points or for the whole orchard fields? Is not clear at all. And for the validation was used any parametric assessment?

164-177 Here, why the yield sampling was excluded and the yield was assessed on vegetation indices. The regression model may predict in better way rather based on yield sampling?

Results

188 The results correspond to validation issue?

255 As was pointed out in above, here the authors must be clarified, why for the analysis was not included the yield sampling, instead of using the vegetation strong indicators only

284 The figure must be replaced by other one

321 Then the discussion chapter must be improved taken into account the all above remarks.

Reviewer 4 Report

Comments and Suggestions for Authors

Dear Authors,

Please find my review attached.

Congratulations on your excellent work — this article shows great potential for publication!

Round 2

Reviewer 2 Report

Comments and Suggestions for Authors

The authors complied with all suggestions. Congrats!

Reviewer 3 Report

Comments and Suggestions for Authors

Dear authors, thanks for improving the manuscript